# Loading Effect of Chitosan Derivative Nanoparticles on Different Antigens and Their Immunomodulatory Activity on Dendritic Cells

**DOI:** 10.3390/md19100536

**Published:** 2021-09-24

**Authors:** Chaojie Xu, Ronge Xing, Song Liu, Yukun Qin, Kecheng Li, Huahua Yu, Pengcheng Li

**Affiliations:** 1CAS and Shandong Province Key Laboratory of Experimental Marine Biology, Center for Ocean Mega-Science, Institute of Oceanology, Chinese Academy of Sciences, Qingdao 266071, China; chaojiexu725@163.com (C.X.); sliu@qdio.ac.cn (S.L.); ykqin@qdio.ac.cn (Y.Q.); lkc@qdio.ac.cn (K.L.); yuhuahua@qdio.ac.cn (H.Y.); 2Laboratory for Marine Drugs and Bioproducts, Pilot National Laboratory for Marine Science and Technology (Qingdao), No. 1 Wenhai Road, Qingdao 266237, China; 3University of Chinese Academy of Sciences, Beijing 100049, China

**Keywords:** chitosan derivative nanoparticles, different antigens, antigen selectivity, encapsulation efficiency, immune effects

## Abstract

Drug carrier nanoparticles (NPs) were prepared by the polyelectrolyte method, with chitosan sulfate, with different substituents and quaternary ammonium chitosan, including C236-HACC NPs, C36-HACC NPs, and C6-HACC NPs. To evaluate whether the NPs are suitable for loading different antigens, we chose bovine serum albumin (BSA), ovalbumin (OVA), and myoglobin (Mb) as model antigens to investigate the encapsulation effect of the NPs. The characteristics (size, potential, and encapsulation efficiency) of the NPs were measured. Moreover, the NPs with higher encapsulation efficiency were selected for the immunological activity research. The results showed that chitosan derivative NPs with different substitution sites had different loading effects on the three antigens, and the encapsulation rate of BSA and OVA was significantly better than that of Mb. Moreover, the NPs encapsulated with different antigens have different immune stimulating abilities to DCS cells, the immune effect of OVA-coated NPs was significantly better than that of BSA-coated NPs and blank NPs, especially C236-HACC-OVA NPs. Furthermore, we found that C236-HACC-OVA NPs could increase the phosphorylation level of intracellular proteins to activate cell pathways. Therefore, C236-HACC NPs are more suitable for the loading of antigens similar to the OVA structure.

## 1. Introduction

Human health is a hotspot of scientific research. The impact of many viruses on human health is complex and difficult to cure. For example, the novel coronavirus in 2020 has killed tens of thousands of people around the world. This virus is extremely contagious, and it is difficult to find a better treatment. Thus, it is necessary to protect uninfected people through preventive measures and minimize the losses caused by the novel coronavirus. Undoubtedly, vaccines are an effective way to treat many infectious diseases. In regard to vaccinia prevention, vaccines are even more important due to their efficacy in preventing diseases. Adjuvants are an essential part of vaccines. The most common adjuvants are white oil and aluminum salts. Both of these adjuvants have some drawbacks [1,2]. Therefore, research on new adjuvants is necessary.

Chitosan exhibits excellent properties in vaccine delivery due to its promising features, such as biocompatibility, biodegradability, and bioadhesion; moreover, it is easily modified by different functional groups [3,4]. Many studies have confirmed that chitosan could stimulate immune cells, such as macrophages [5,6], and dendritic cells [7,8,9], to exert immune effects. Furthermore, chitosan derivatives, such as 2-hydroxypropyl trimethyl ammonium chloride chitosan (HACC) [10] and N, N, N-trimethyl chitosan [11], have important immune activity. This indicates that chitosan and its derivatives have potential as immunostimulants. An adjuvant could better exert the immunostimulatory effect of chitosan. Chitosan can also be biodegraded in the human body.

In recent years, chitosan and its derivative NPs have received widespread attention. This may be related to several advantages, including enhancing the uptake of antigen presenting cells, protecting antigens from enzymatic degradation, and promoting the depot effect through the gradual release of antigen and immunomodulators to the same cell population [12,13]. Nanoparticles can convert soluble antigens into particulate antigens, which has better immunogenicity. Chitosan-aluminum NPs can better induce cellular and mucosal immune responses and show higher IgG secretion, compared to aluminum salt alone [14]. Compared to the Newcastle disease vaccine, the HACC NP-loaded Newcastle disease vaccine could enhance the immune response [15].

In addition, due to the different structure of antigens, not every kind of nanocarrier is suitable for all kinds of antigens. Proteins with different isoelectric points bind to NPs in different ways. For example, the NPs prepared by the TPP cross-linking method can better adsorb BSA with a low isoelectric point [16]; whereas the loading effect of proteins with different charges, such as OVA, BSA, lactalbumin, and casein, are quite different [14]. A study showed that OVA had a significantly higher ability to bind chitosan-C48/80 NPs than myoglobin [17].

Moreover, in our previous study, we demonstrated that the NPs prepared with sulfated chitosan (SCS) and 2-hydroxypropyl trimethyl ammonium chloride chitosan (HACC) have immunostimulatory effects in DCS cells [18]. Screening from molecular weight and substitution sites, it is determined that chitosan sulfate derivatives with a molecular weight of 200 kDa and three substitution sites (C236, C36, and C6) have effective immunological activity. In order to determine which antigen is more suitable for the loaded optimized chitosan derivative NPs, in this study, three different antigens (BSA, OVA, and Mb) with different isoelectric points were used as model antigens to study the loading effect of chitosan derivative nanoparticles. After screening the antigen with the best loading effect, the immune activity of the antigen loaded NPs and its effect on the cell pathway were studied. That was very important for a nanocarrier as an adjuvant.

## 2. Results

### 2.1. Characterization of Chitosan Derivatives

The derivatives were prepared according to the method described in our previous study [18]. FTIR and NMR (^1^H NMR and ^13^C NMR) were used to characterize the derivatives. The results indicate that all derivatives were successfully prepared [18].

### 2.2. Characterization of the NPs

Nanoparticles with three different antigens were prepared, and their potentials were 16.00–38.40 mV, particle sizes 163.20–255.00 nm, and polydispersity index (PDI) 0.079–0.336 (Table 1.). Some of the data about the antigen OVA were recorded in our previous study [18].

After measuring the particle size and potential of the NPs with a nanoparticle size analyzer, we used the transmission electron microscope (TEM) to characterize the morphology of the NPs, to ensure that the morphology of the NPs was uniform and consistent with the measurement results of the particle size analyzer (Figure 1).

### 2.3. Encapsulation Efficiency of NPs

The encapsulation efficiencies of NPs for three antigens are shown in Figure 2. Different types of NPs have different encapsulation efficiencies for the three antigens. Among the three types of nanoparticles, BSA has the highest encapsulation rate, and Mb has the lowest encapsulation rate. The entrapment rate of BSA by chitosan derivative nanoparticles with different substitution sites was more than 75%. For OVA, the entrapment rate of C36-HACC for OVA was slightly lower than 70%, and the other two were higher than 70%, but the entrapment rate of Mb by three kinds of nanoparticles was lower than 20%. Due to the low encapsulation efficiency of the nanoparticles to Mb, it is difficult to be used as a model antigen of the nanoparticles for subsequent experiments. Therefore, no subsequent determinations included samples loaded with Mb.

### 2.4. Cytotoxicity of NPs

CCK-8 was used to determine the survival rate of DCS cells under different concentrations of NPs. The results showed that (Figure 3.), for blank NPs, DCS cells are non-toxic within 250 μg/mL. NPs with two different antigens have different cytotoxicity to cells, but all NPs are non-toxic within 250 μg/mL for DCS cells.

### 2.5. RNA Expression of NPs on DCS Cells

Different NPs have different expression levels of immune factors on DCS cells. As shown in Figure 4, the same NPs have different expression levels of different cytokines. The gene expression levels of IL-6, TNF-α, and IL-1β of OVA loaded with three kinds of NPs were significantly higher than those of BSA-loaded NPs and blank NPs. Compared with the blank NPs, the BSA-loaded NPs increased the expression of IL-6, and the BSA-loaded NPs with different substitution sites had no effect on the TNF- α and IL-1β expression levels. For example, compared with C236-HACC, C236-HACC-BSA decreased the TNF-α expression level, and compared with C36-HACC and C6-HACC, C36-HACC-BSA and C6-HACC-BSA increased the TNF-α expression levels. In general, OVA-loaded NPs have the best effect on promoting immune factors, especially C236-HACC-OVA NPs.

### 2.6. The Protein Phosphorylation Level in Cell Pathway

According to the results, the NP-loaded OVA can enhance the expression of some genes (IL-6, TNF-α, and IL-1β) that have immune effect. To further explore the mechanisms of gene expression in DCS cells by NP-loaded OVA antigen, the cells were treated with NPs in a concentration of 100 μg/mL.

The result was shown in Figure 5, compared with the DMEM group (blank control), the phosphorylation level of the protein in the cells treated with the NPs had increased, indicating that the NPs can activate the intracellular protein pathways, then exert an immune effect. C236-HACC-OVA NPs can increase the AKT-1, PDK1, ERK, JNK, and p38 phosphorylation level compared to C6-HACC-OVA NPs, which indicated that C236-HACC-OVA NPs could better activate the PI3K-Akt pathway in dendritic cells. This indicates that NPs promote nuclear transcription factors to enter the nucleus, thereby promoting the expression of the immunocompetent gene.

### 2.7. The Determination of NPs Endotoxin

The endotoxin content was determined by the endotoxin detection kit. The distilled water was used as a negative control and LPS was used as a positive control. As shown in Table 2, the measurement results showed that the endotoxin of C236-HACC-OVA NPs was 0.10 EU/mL, and the endotoxin of C6-HACC-OVA NPs was 0.29 EU/mL. This result ruled out the impact of endotoxins on cells and proved the immunostimulatory properties of the nanoparticles.

### 2.8. The Release of NPs

The results show that, with the increase of time, the OVA in the NPs is gradually released. Among them (Figure 6), the release rate is the fastest at 60 h; at 146 h, it basically reaches the highest point of release. This indicates that the NPs can be released from NPs almost completely in one week. This laid the foundation for our follow-up animal experiment.

## 3. Discussion

The loading effect of the same nanoparticle for different antigens is different, indicating that the nanoparticle may have a recognition effect on the structure of the antigen [17]. Only the antigen with the same or similar type of structure can be better loaded by the nanoparticle. This shows that nanoparticles can be used as a qualified adjuvant.

In our study, BSA had the highest encapsulation efficiency, and Mb had the lowest encapsulation efficiency. The encapsulation efficiencies of BSA by chitosan derivative NPs with different substitution sites were more than 75%. For OVA, the encapsulation efficiencies of NPs were about 70%, but the encapsulation efficiencies of Mb by three kinds of NPs were lower than 20%. The pI of OVA and BSA were both less than 7, and the pI of Mb was greater than 7. Under neutral conditions, the prepared NPs were at pH = 7. Thus, OVA and BSA showed higher encapsulation efficiency. This is consistent with the conclusion in the previous study [14,19]. On the other hand, although those three proteins are standard antigens, they have different molecular weights (Mw). The molecular weight of OVA is 45 kDa, BSA is 68 kDa, and Mb is 17 kDa. This proves that the loading effect is also closely related to the molecular weight of the protein [17].

The same antigen was loaded by NPs prepared from different chitosan derivatives; their immunological effects are different. Although the encapsulation efficiency of OVA and BSA were high, the immune level of the NPs loading OVA and BSA was different. This indicates that the immune level is not only connected to the encapsulation efficiency but is also related to the structure of the antigen [20,21]. The protein adsorption and release are complicated processes, which are closely related to the structural characteristics of the protein [17]. The results show that C236-HACC NPs are more suitable for antigens with pI lower than 7 and a protein structure similar to OVA.

After the cells were treated by different NPs for 12 h, the protein phosphorylation level in the cell increased. This corresponds to the increase in the expression of cellular immune factors by the NPs, indicating that the increase in the phosphorylation level of the cell protein activates the cellular pathway and stimulates the cell to produce immunity. This can explain the response to secrete cytokines. This is consistent with many previous studies. The immune level enhancement would increase the phosphorylation level of intracellular pathway proteins [5,22].

Different types of NP-loaded OVA and immune activity are higher than the NP-loaded BSA. In different types of NP-loaded OVA, when C236-HACC-OVA and C6-HACC-OVA act to the cells, they have different effects on the level of intracellular protein phosphorylation. Previous research documents have also shown that different substitution sites and different degrees of substitution also have a greater impact on the activity of chitosan derivatives [23,24]. This shows that the activity of the derivatives affect the activity of the NPs. We subsequently conducted animal (BALB/C mice) experiments. Through the determination of stability and release of the NPs, the interval between two immunizations during the animal experiment was one week. This is consistent with the release of antigen in our nanoparticles.

## 4. Materials and Methods

### 4.1. Materials

Chitosan (MW = 200 kDa) was prepared by our laboratory, ovalbumin (OVA) was bought from Sigma-Aldrich (Saint Louis, MO, USA), myoglobin (Mb) and bovine serum albumin (BSA) were bought from Solarbio, N, N-dimethylformamide, formamide, N-methyl-2-pyrrolidone, chlorosulfonic acid, phthalic anhydride, and ethylene glycol came from Sinopharm Chemical Reagent Co. Ltd. (Shanghai, China). DMEM medium and penicillin–streptomycin came from HyClone (Logan, Utah, USA). Fetal bovine serum (FBS) was acquired from Gibco in Australia; PrimeScript RT reagent Kit (including gDNA Eraser) and SYBR Premix Ex Taq II (Tli RNase H Plus) were bought from Takla in Shiga, Japan. CCK-8 was supplied by ApexBio (Houston, TX, USA), E.Z.N.A. Total RNA Kit was bought by OMEGA Biotechnology Company in, Norcross, GA, USA; ToxinSensor™ Chromogenic LAL Endotoxin Assay Kit was bought from GenScript (Piscatway, NJ, USA).

### 4.2. Preparation of Hydroxypropyl Trimethyl Ammonium Chloride Chitosan (HACC) and Chitosan Sulfate (SCS)

Chitosan derivatives (HACC, C236-SCS, C36-SCS, and C6-SCS) were synthesized, as we previously reported [18]. The methods were based on Yang [5] and Xing [25]. For HACC, 5 g chitosan was mixed with 10 g of 2,3-epoxypropyltrimethylammonium chloride and 80 mL of distilled water was added to it; mechanical stirring lasted for 24 h at 85 °C.

C2,3,6 chitosan sulfate was prepared by the method by Yang [5]. The chitosan powder (2 g) was introduced in a three-necked flask containing 50 mL formamide and 5 mL formic acid, with swirling, to get gelatinous chitosan. The resulting mixture was precipitated by 3 times the volume of absolute ethanol and placed in a refrigerator at 4 °C for about 30 min. The precipitate was then filtered through a Buchner funnel under reduced pressure and washed with EtOH. The precipitate was dissolved in distilled water and was then neutralized with 2M NaOH solution.

C3,6 and C6 chitosan sulfate were prepared by the method by Xing [20]. For C3,6 chitosan sulfate, briefly, 4 g chitosan was added in 100 mL DMF, with stirring, then 5 g phthalic anhydride and 3 mL ethylene glycol were added to the reaction system in turn. The mixture reacted at 90 °C for 2.5 h with stirring. The reaction system was then cooled to 55 °C and 80 mL DMF·SO3 was added. After reacting for 2.5 h, the reaction solution was poured into ice water and neutralized with 2 M NaOH to obtain a transparent solution. A total of 3 g 2-phthalimidochitosan sulfate was fully dissolved in 100 mL distilled water and 20 mL hydrazine hydrate reacting for 4 h at 70 °C. Moreover, 100 mL of distilled water was added to the reaction system and concentrated under reduced pressure (for 4 times).

For C6 chitosan sulfate, 3 g chitosan was dissolved in 80 mL of 1% formic acid in a beaker. The chitosan powder (3 g) was introduced in a three-necked bottomed flask containing 1% formic acid with mechanical agitation to mix evenly. A total of 4.65 g of CuSO4 5H2O was dissolved to 10 mL distilled water and then dropped to the above-mentioned solution at RT. The reaction lasted for 3 h at room temperature and stirring continued for 3 h after adjusting the pH to 6.0–6.5 with ammonia water. Afterward, the resulting precipitate was filtered. Next, 2 g chitosan copper chelate, 50 mL formamide, and 30 mL DMF·SO3 reacted at 55 °C for 1.5 h. The mixture was then precipitated with 3 times the volume of anhydrous ethanol and placed in a refrigerator at 4 °C for 30 min. The product was filtered and dissolved in distilled water. Afterward, the solution was scattered in distilled water, using a strongly acidic styrene-based cation exchange resin column. The solution neutralized with 2 M NaOH.

All chitosan derivatives were obtained after dialysis, concentration, and freeze-drying.

### 4.3. Preparation of Chitosan Derivative NPs Loaded with Different Antigens

Three solutions consisting of 5 mg/mL of positively charged solution (HACC solution), 1 mg/mL chitosan sulfate solution (C236, C36, and C6 solution), and 2 mg/mL antigen solution (BSA, OVA, and Mb solution) were prepared. Moreover, 5 mL HACC solution were introduced into a beaker, magnetically stirring at 600 r/min at 25 °C, and 2 mL antigen solution were added dropwise, then 2 mL chitosan sulfate solution were added dropwise, stirring continued for 30 min. NP solution was obtained after filtration. Finally, the NPs loaded with antigen were obtained and stored at 4 °C.

### 4.4. Characterization of Different Chitosan Derivatives

The prepared HACC and SCS were characterized by Fourier transform infrared (FT-IR) and nuclear magnetic resonance (NMR) respectively. The sample preparation method and measurement method were the same as we previously reported [18].

### 4.5. Characterization of Chitosan Derivative NPs Loaded with Different Antigens

The prepared NPs are characterized using potential and particle size by Zetasizer (ZS90, Malvern Instruments, Malvern, UK), as in our previous study [18].

### 4.6. Encapsulation Effect of NPs

The antigen could be wrapped by NPs—this is the first property for NPs to be used as the adjuvant. Therefore, we determined the encapsulation efficiencies of the NPs in three different antigens to test the NPs, selective for different antigens. The prepared NPs were centrifuged at 12,000 × *g*/min for 30 min in 4 °C. Afterward, we determined the protein content *N* in the supernatant. At the same time, the protein content *M* in the same concentration solution, containing only the antigen protein, was determined.

The encapsulation efficiency (EE) was calculated by the following equation:
(1)$$\mathrm{EE}\%\;=\;\frac{M-N}{M}\;\times \;100$$

### 4.7. Cell Culture

The DCS cells were obtained from the National Infrastructure of Cell Line Resource (Beijing, China). DCS cells were maintained in DMEM medium (including 10% FBS and 1% antibiotics) with 5% CO_2_ incubator at 37 °C.

### 4.8. In Vitro Experiment

#### 4.8.1. Cytotoxicity of NPs

To ensure that the NPs had good biocompatibility and non-toxicity, the CCK-8 kit was used to determine the cytotoxicity of the NPs. First, DCS cells were inoculated into a 96-well plate. After 12 h, the medium was changed to new medium with NPs; 24 h later, 10 μL of CCK-8 solution was added to every well and incubated in a 37 °C incubator for 2 h. Finally, a spectrophotometer was used to measure the absorbance at 450 nm.

#### 4.8.2. RNA Expression Analysis

An OMEGA kit was used to extract RNA, and then a Takara kit was used to remove cDNA and reverse RNA. Briefly, DCS cells were incubated at 37 °C for 12 h, then the medium was changed by the NPs (100 μg/mL). After 24 h, the 6-well plate was taken out, washed twice with PBS, and then GTC extract was added to lyse the cells. In the meantime, ethanol was added to homogenize the lysate. Afterward, the mixture was put on the RNA Collection Column and then the columns were centrifuged at 12,000× *g* for 1 min to obtain a crude RNA extract. Finally, the RNA product was obtained after washing several times.

The reaction was performed to remove gDNA and reverse transcription. Next, after adding the forward primer, reverse primer (Table 3.), and reagents, the RT-PCR reaction was carried out. The reaction procedure is as follows: Stage 1, 95 °C 30 s; Stage 2, 95 °C 5 s–60 °C 34 s; Stage 3, 95 °C 15 s–60 °C 60 s–95 °C 15 s.

#### 4.8.3. Determination of Protein Phosphorylation Level in Cell Pathway

DCS cells were culture in a 6-well plate after digesting; 12 h later, the medium was changed to a medium containing 100 μg/mL NPs. After 24 h, the cells were washed twice by PBS; the cells were then scraped with a cell scraper and resuspended with PBS. Then the PBS with cells were centrifuged at 1200× *g*/min for 5 min, and the PBS was then removed. Afterward, a 200 μL lysis buffer and a Roche Complete protease inhibitor cocktail were introduced to extract total proteins, and were centrifuged at 12,000× *g*/min for 5 min. We collected the supernatant for analysis. Then the protein separation kit was used to extract the protein. The BCA kit was used to determine the content of the protein.

For the detection of specific proteins, 40 μg of protein was denatured and separated by 8–15% sodium dodecyl sulfate-polyacrylamide gel electrophoresis. Then, we transferred the proteins to polyvinylidene fluoride membranes. Next, the membranes were washed with 10 mL TBS. After 10 min, 5 mL BSA were added to block the nonspecific binding. The membranes, after being rinsed by TTBS (three times), were incubated with corresponding specific primary antibodies and horseradish peroxidase-conjugated secondary antibodies for 1 h. The bounded antibodies were imaged with the Tanon-5200 chemiluminescence detection system (Tanon Science, Shanghai, China) after being washed by TTBS (three times).

### 4.9. The Endotoxin of NPs

The application of nanoparticle as an adjuvant must first eliminate the impact of NP endotoxin. Thus, we used the ToxinSensor™ Chromogenic LAL Endotoxin Assay Kit to determine the endotoxin of NPs. According to the operation steps of the kit, first, we configured the required solutions, LAL solution, chromogenic substrate solution, LAL reagent water, color stabilizer no. 1, color stabilizer no. 2, and color stabilizer no. 3. The standard and sample solutions were then taken, 100 μL, respectively, and added to the sample bottles in turn. After mixing, the bottles were placed in a 37 °C water bath for 10 min. After taking them out, 100 μL of chromogenic substrate was added to each bottle. After 6 min at 37 °C, color stabilizer no. 1, color stabilizer no. 2, and color stabilizer no. 3 were added to each bottle in turn. Finally, we measured the absorbance at 545 nm and calculated the endotoxin value.

### 4.10. The Antigen Release of NPs

We took out the prepared NP-loaded OVA in distilled water and placed on a shaker at 37 °C. Then we sampled 200 μL of NP solution every few hours, centrifugated at 12,000× *g* for 30 min. The protein content in the supernatant was measured using the BCA kit, and 2 mg/mL OVA was measured as a control.

## 5. Conclusions

To develop an effective vaccine adjuvant, one of the most important metrics is the loading effect of the antigen. This is the first important part to measure whether the nanoparticles can be used as a suitable carrier. Our results show that C236-HACC-OVA NPs have high encapsulation efficiency and strong immunostimulatory activity. This shows that the nanoparticles prepared by our chitosan derivatives have a better encapsulation rate for OVA-structured antigens, and it can subsequently be used as a carrier and adjuvant for this type of antigen.

Once we consider the NPs (prepared by chitosan) to use as a vaccine adjuvant, it is necessary to select suitable antigens for the adjuvant in order to exert the best effect of the vaccine. Through the preliminary exploration of intracellular pathways, it is verified that nanoparticles can activate cellular pathways. Thus, NPs can cause an immune response, which lays the foundation of further exploration for the mechanism action of NPs.

## Figures and Tables

**Figure 1 marinedrugs-19-00536-f001:**
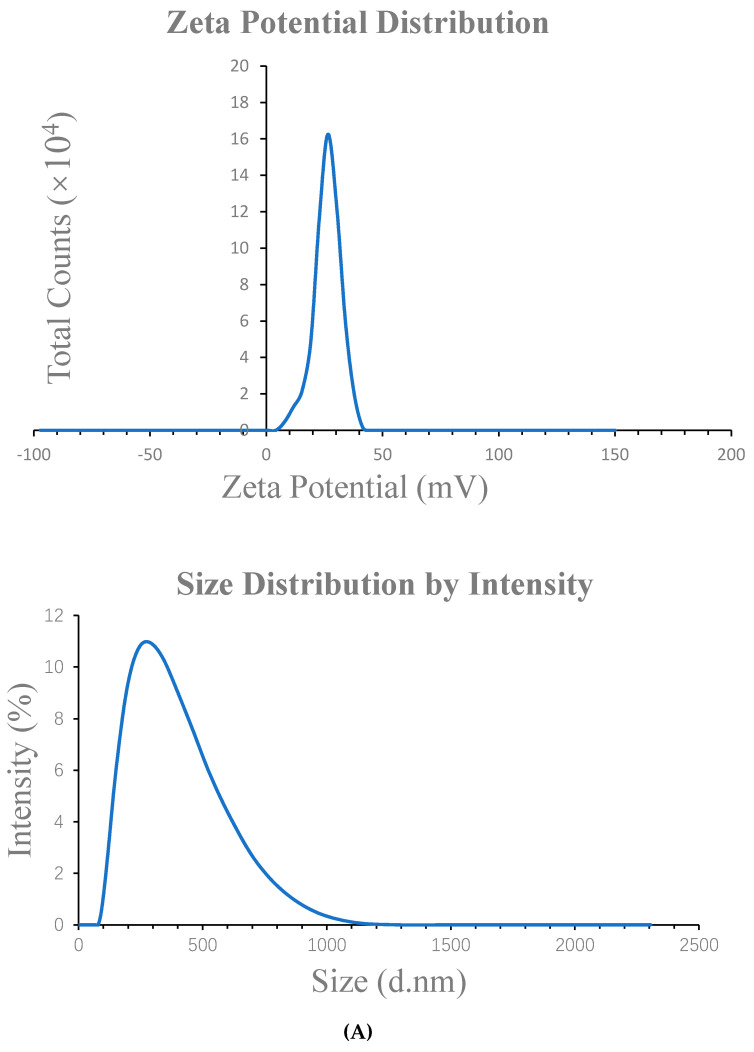
(**A**) The zeta potential average and diameter of the C236-HACC-OVA NPs. (**B**) The zeta potential average and diameter of the C236-HACC-BSA NPs. (**C**) The TEM images of NPs at 2 μm sizes of the C236-HACC-OVA NPs and C236-HACC-BSA NPs.

**Figure 2 marinedrugs-19-00536-f002:**
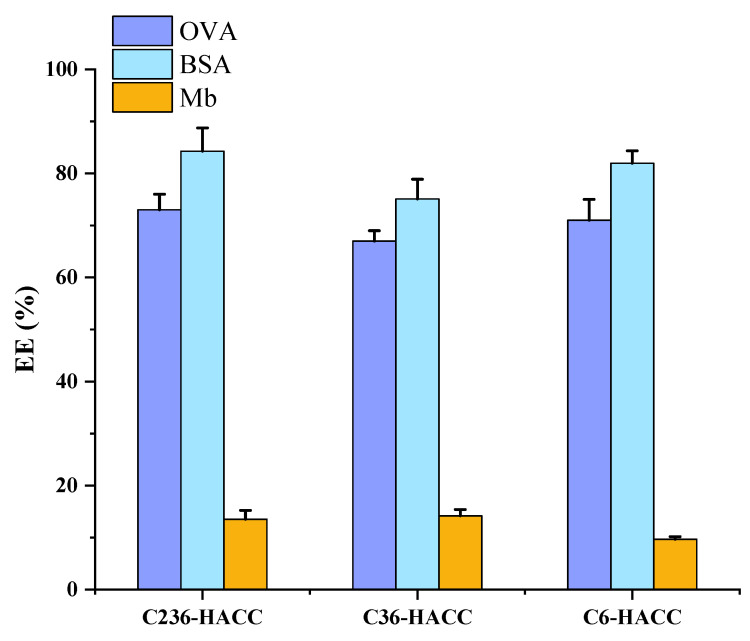
The encapsulation efficiencies of nanoparticles for three antigens (OVA, BSA, and Mb). The data are presented as mean ± standard deviation (*n* = 3).

**Figure 3 marinedrugs-19-00536-f003:**
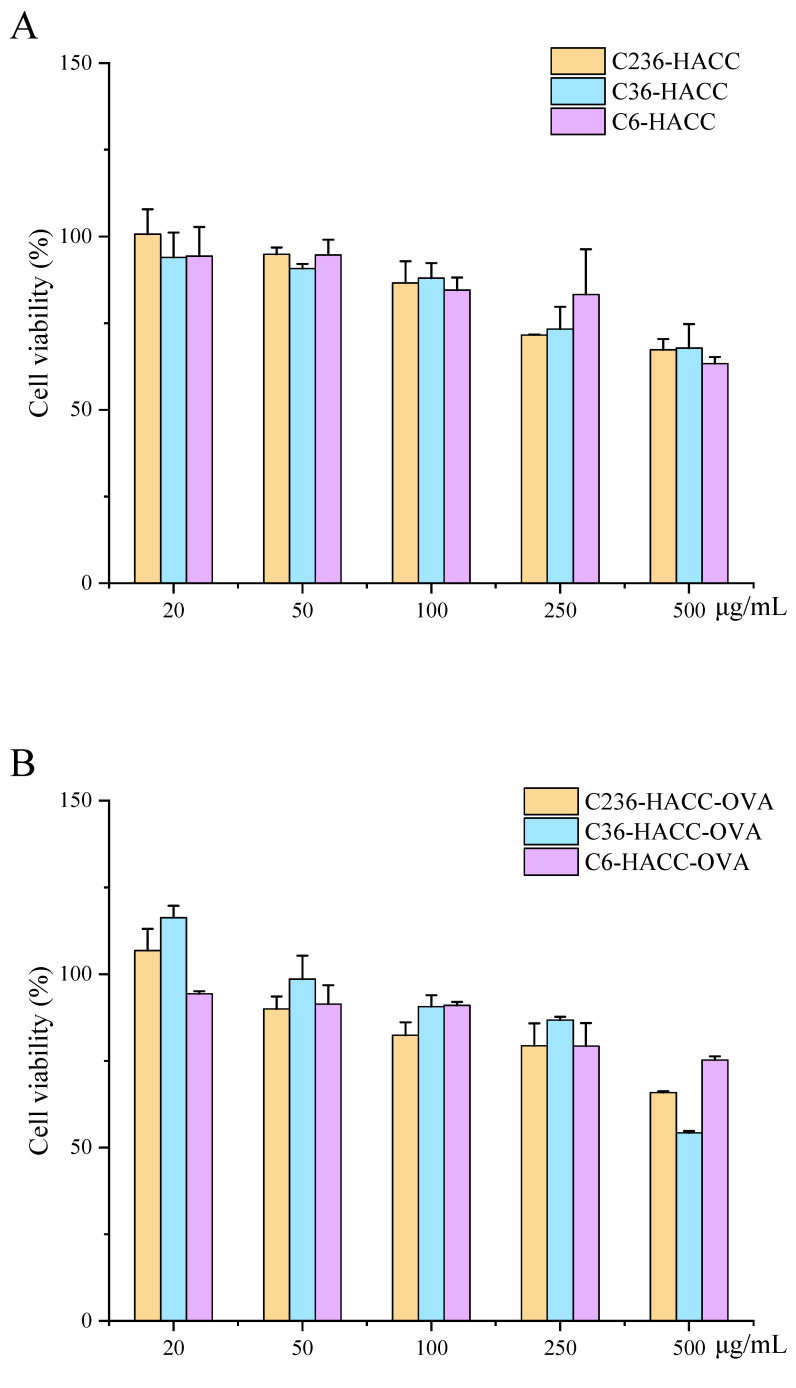
The cell viability for (**A**): blank nanoparticles (NPs); (**B**): NP-loaded ovalbumin (OVA); (**C**): NP-loaded bovine serum albumin (BSA). The data are presented as mean ± standard deviation (*n* = 3).

**Figure 4 marinedrugs-19-00536-f004:**
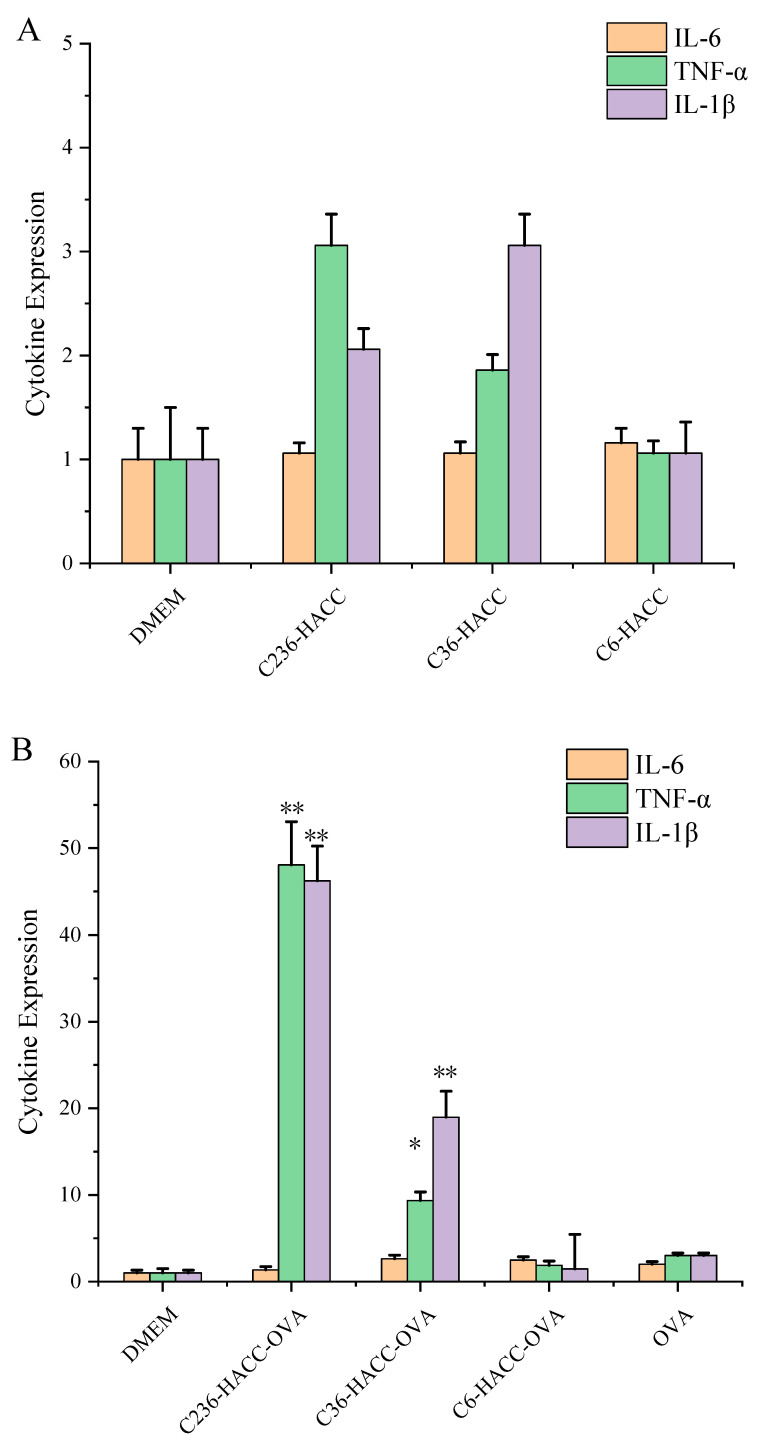
Gene expression was measured by RT-PCR. (**A**) Interleukin (IL)-6, tumor necrosis factor-α (TNF-α), and IL-1β expression in DCS cells stimulated by blank NPs; (**B**) IL-6, TNF-α, and IL-1β expression in DCS cells stimulated by NP-loaded ovalbumin (OVA) antigen; (**C**) IL-6, TNF-α, and IL-1β expression in DCS cells stimulated by NP-loaded bovine serum albumin (BSA) antigen. The data are presented as mean ± standard deviation (*n* = 3). NPs versus OVA: * *p* < 0.05, ** *p* < 0.01.

**Figure 5 marinedrugs-19-00536-f005:**
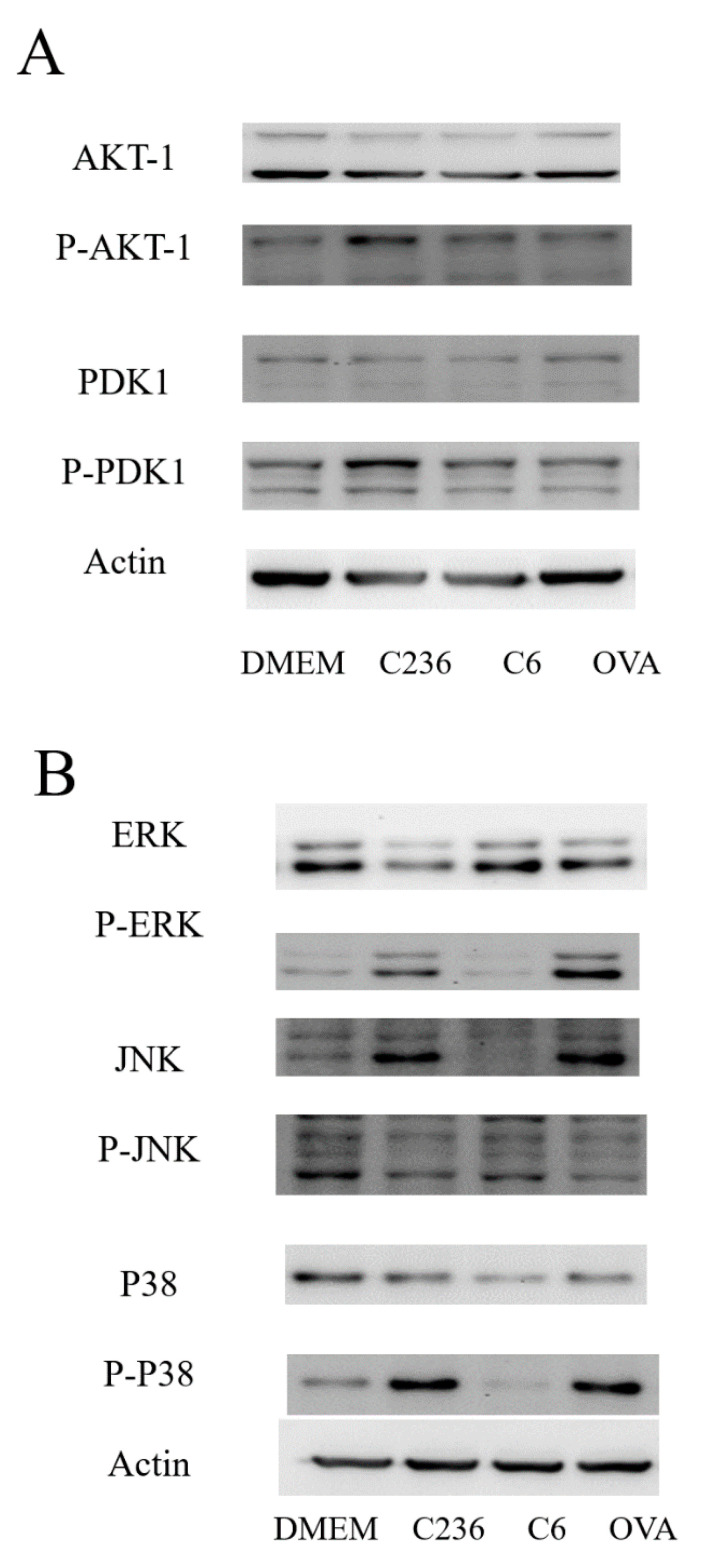
Determination of C236-HACC-OVA NPs (C236) and C6-HACC-OVA NPs (C6) induced signaling pathways. DCS cells were stimulated with 100 μg/mL of NPs for 12 h, and then the cell proteins were extracted and subjected to western blotting. PI3K-Akt (**A**) MAPK (**B**), signaling pathways were determined.

**Figure 6 marinedrugs-19-00536-f006:**
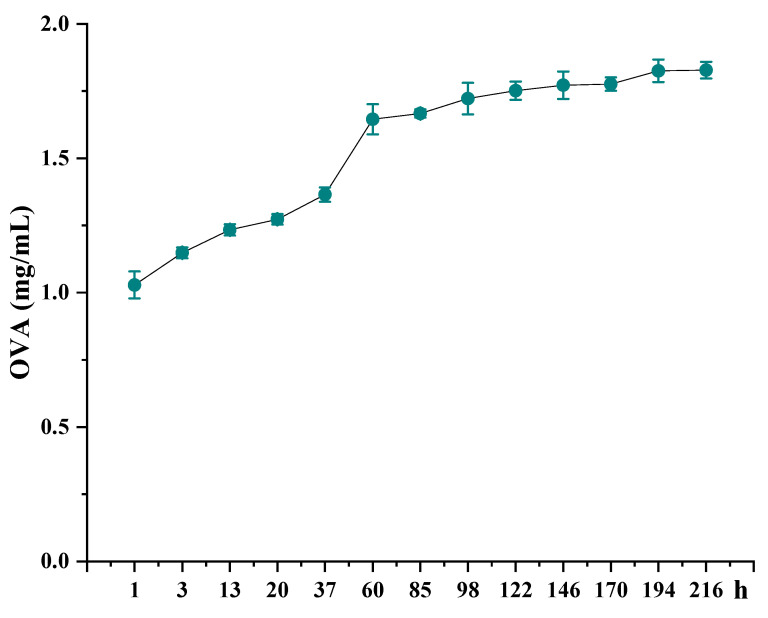
The release of C236-HACC-OVA NPs on a shaker at 37 °C. The data are presented as mean ± standard deviation (*n* = 3).

**Table 1 marinedrugs-19-00536-t001:** The particle size average, zeta potential, and polydispersity index (PDI) of nanoparticles with positive charge.

Sample	Antigen	Zeta Potential Average (mV)	Size Average (nm)	Polydispersity Index
C236-HACC	OVA	25.40 ± 1.80	248.50 ± 21.56	0.173 ± 0.050
	BSA	38.40 ± 2.11	163.20 ± 19.87	0.187 ± 0.009
	Mb	38.40 ± 0.99	213.40 ± 15.43	0.079 ± 0.013
C36-HACC	OVA	30.50 ± 0.81	210.80 ± 12.76	0.143 ± 0.061
	BSA	31.30 ± 2.41	193.30 ± 17.54	0.204 ± 0.018
	Mb	16.00 ± 0.21	255.00 ± 15.24	0.336 ± 0.016
C6-HACC	OVA	28.33 ± 3.67	250.80 ± 6.71	0.188 ± 0.001
	BSA	32.90 ± 2.40	135.70 ± 31.81	0.191 ± 0.054
	Mb	32.80 ± 1.98	217.30 ± 29.10	0.153 ± 0.003

**Table 2 marinedrugs-19-00536-t002:** Endotoxin determination of two kinds of NPs, control and LPS (EU/mL).

Control	C236-HACC-OVA NPs	C6-HACC-OVA NPs	LPS
0.0096	0.10 ± 0.009	0.29 ± 0.011	0.63 ± 0.021

**Table 3 marinedrugs-19-00536-t003:** Real-time PCR primer sequences (5′ to 3′).

Primer	Forward Primer Sequences	Reverse Primer Sequences
IL-6	TGGGACTGATGCTGGTGACA	ACAGGTCTGTTGGGAGTGGT
IL-1β	GCAGAGCACAAGCCTGTCTTCC	ACCTGTCTTGGCCGAGGACTAAG
TNF-α	GCGACGTGGAACTGGCAGAAG	GCCACAAGCAGGAATGAGAAGAGG
GAPDH	ACTCACGGCAAATTCAACGGCA	GACTCCACGACATACTCAGCAC

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
