# Peer review of "Loading Effect of Chitosan Derivative Nanoparticles on Different Antigens and Their Immunomodulatory Activity on Dendritic Cells"

_marinedrugs, 2021, doi:10.3390/md19100536_

Round 1

Reviewer 1 Report

Journal: MDPI – Marine Drugs

Manuscript

Title:  " Loading Effect of Chitosan Derivative Nanoparticles on Different Antigens and Their Immunomodulatory Activity on Dendritic Cells"

Author(s): Chaojie Xu, Ronge Xing, Song Liu, Yukun Qin, Kecheng Li, Huahua Yu,

 and Pengcheng Li

Reviewer Comments to Author(s)

Recommendation: Major revisions

In this articles, the author(s) present a very interesting research work on antigens loading in chitosan NPs and their immunomodulatory activity. This study over antigen-loaded chitosan nanoparticles comes in continuation of the previous research study concerning chitosan derivatives (SCS and HACC) potential immunostimulatory effects (reference 18). However, major concerns need to be considered (line 59, line 152. Grammatical and syntactic errors need to be revised throughout the manuscript.

The author(s) might consider the following:

  1. Throughout the manuscript author(s) are referred to protein (BSA, OVA, Mb) entrapment, or to encapsulated, or coated NPs. These are different terms. Moreover, the same characterization should be followed throughout the manuscript. It is highly encouraged to provide a scheme explaining/visualizing the entrapment/or absorption of proteins in the polyelectrolytic NPs and the structure of the NPs themselves. It is not clear if proteins are entrapped or absorbed and where.
  2. Scale bars should be provided within the TEM images of Figure 2 C, and magnification need to be mentioned.
  3. discussion part could be strengthened in terms of structure. The results are shortly discussed, and the figures/tables to which they correspond are not mentioned. Also, for such important aspects that are analyzed, the results could be more connected to the literature. For that and in order to better document the results, the authors are encouraged to enrich the part of discussion.
  4. Increased entrapment rate of proteins is highly interesting and especially of large proteins such as BSA with 583 amino acid residues and OVA with 385 residues. The significantly lower entrapment rate of Mb is attributed to its isoelectric point being greater than 7 and the loading process followed at pH 7. However, it is not of doubt that Mb is the smallest of the three proteins with only 153 amino acid residues and almost 4 time less encapsulation rate than BSA and OVA. Which other characteristics may also play crucial role for such a low entrapment? The author(s) are encouraged to provide some explanations and references (if needed) on the subject within discussion.
  5. Line 205: How can the author(s) be certain that NP act on the inside of the cells? Have they performed intracellular characterization to rate or visualize the fate of the NPs inside the cells? Otherwise, they can not possible know the uptake of the NPs by the cells but only the internalization of the antigens. For example, OVA or BSA since are entrapped they could have been internalized either by the internalized NPs and released inside the cell or released outside the cell and then internalized or both. However, the author(s) have no evidences on NPs internalization. Thus, this conclusion should be revised unless author(s) have results to provide on NPs internalization. Moreover, the release profile of the proteins has been studied by the author(s)? Since the cells are treated with protein-loaded NPS, it would be useful to have a sense of the release profile of the proteins at the same time period. The antigens definitely act intracellularly but with no evidence of NPs entering the cells (visualized or in which portion), the proteins’ release profile would add value to the time scale of protein release – internalization.
  6. The section of In vitro Experiment (4.8) seems to be written in a hurry and presents many grammatical / syntactic errors and omissions. 1. Culture protocol of DCS cells is missing. 2. in paragraph 4.8.2: line 303 RNA not RAN, line 304-5 the cells medium is mentioned that was changed by the NPs. In which concentration and in which medium the NPs were? line 306-307 " anhydrous was ... lysate", line 307 "was puts", line 308 "centrifuges" here are grammatical and syntactic errors through all the sentences. Moreover, important aspects of the PCR protocol are missing, such as list of primer sequences, PCR process that was followed, Takara kit (that is PCR Amplification Kit with Takara Taq) has not been listed in Materials (4.1). In their previous work (ref.18) the author(s) according to the Materials section (4.1 here, §2.1 at ref.18) use the same kits for RNA Analysis and probably the same processes, but their work is not referenced here, so as to find protocol details. Detailed protocols should be either mentioned or referenced in sections 4.7 and 4.8

Reviewer 2 Report

The paper Loading Effect of Chitosan Derivative Nanoparticles on Different Antigens and Their Immunomodulatory Activity on Dendritic Cells from Chaojie Xu, Ronge Xing, Song Liu, Yukun Qin, Kecheng Li, Huahua Yu and Pengcheng Li, is seems interesting in the area chitosan nanoparticles as drug carriers.

This paper can not be accepted in this form because most of the text is copied from the article of the same authors: Immunostimulatory effect of N-2-hydroxypropyltrimethyl ammonium chloride chitosan-sulfate chitosan complex nanoparticles on dendritic cells. Although this previous paper is refereed in ref. [18] however they use the same text (words, numbers, …) and this is a symptom of plagiarism. This is also observed in Figure 1 which spectra are the same as in ref [18].

Besides there are a lot of questions that must be taken into account. For example in 4.4. characterization of Different Chitosan Derivatives, the authors describe two FTIR instuments but they only use one for spectra of Figure 1. At the same time they describe the NMR instrument but they don´t show any spectra in the paper (probably this is copied from other papers and mainly in ref [18]).

The Mb solution is not defined nor described in the text and then results of Figure 3 can not be correctly analyzed. Similarly is for DMEM, Actin, P-PDK1, etc. of Figure 6.

The English language is difficult to follow in some paragraphs.

Reviewer 3 Report

The manuscript entitled “Loading Effect of Chitosan Derivative Nanoparticles on Different Antigens and Their Immunomodulatory Activity on Dendritic Cells” by Xu and coworkers aim to prepared different chitosan-based nanoparticels to stimulate dendritic cells. Most of the data appear very preliminary. This prevents clear conclusions. Important experiments such as NP uptake by the cells or NP stability in culture medium are missing and/or are not discussed.

  1. The authors characterized their NPs by different methods. However they showed no standard deviation/error bars (Table 1 or Figure 3) and they did not mentioned how often they repeated their experiments? Please provide the missing information.
  2. The authors should provide data on the stability of the NPs in culture medium.
  3. The endotoxin levels in the prepared NPs are relatively high. Again, no standard deviation is indicated. This is important in order to understand the reliability of the cytokine expression data.
  4. The discussion is too scarce. The authors have to explain why encapsulation of e.g. OVA increase the levels of cytokine expression in die DCs. Why does OVS have immunogenic properties once it is encapsulated? Is this related to particle uptake by the DCs?
  5. I suggest that the authors carefully revise the language of the manuscript. For instance, the sentence in the abstract “In order to evaluate whether the carrier is suit-14 able for loading different antigens, bovine serum albumin (BSA), ovalbumin (OVA) and myoglobin 15 (Mb) with different isoelectric points were used as model antigens to investigate the encapsulation 16 effect of the prepared nanoparticles on them.” is difficult to understand.

Round 2

Reviewer 1 Report

Journal: MDPI – Marine Drugs

Manuscript

Title:  " Loading Effect of Chitosan Derivative Nanoparticles on Different Antigens and Their Immunomodulatory Activity on Dendritic Cells"

Author(s): Chaojie Xu, Ronge Xing, Song Liu, Yukun Qin, Kecheng Li, Huahua Yu,

 and Pengcheng Li

Reviewer Comments to Author(s)

Recommendation: Accept

After a detailed evaluation of the revised manuscript all the issues and questions have been addressed and the manuscript can be accepted for publication.

Author Response

Thank you very much for giving us the opportunity to improve the quality of the manuscript, and we will conduct more in-depth exploration in subsequent experiments.

Reviewer 2 Report

The paper can be accepted but the Figure 1 must be reduced by plotting the two particle size distributions in only one plot and the two Zeta Potentials in only one plot.

Author Response

(The authors gave the same response as above.)

Reviewer 3 Report

The authors addressed some of the raised points. Now, they mentioned mostly the number of experiments and they stated their data with standard deviations. However, several aspects appear still preliminary and also the discussion suffers from speculations and lack of references.

  1. The authors included a new diagram showing the release of OVA. Again, no standard deviation is shown. What kind of NPs were tested? Why only OVA and not BSA and Mb? How is this experiment related to the biological effects the author measured? Is the release important for the biological effect? It would be much better to show the uptake of the NPs into the cells.
  2. The authors wrote: “This indicates that the NPs can be released almost completely in one week.” Did they mean “This indicates that OVA can be released from NPs….”.
  3. The authors mentioned animal experiments in the results section? “This laid the foundation for our follow-up animal experiment.” It is better to state future experiments in the discussion or the conclusion.
  4. The discussion contains a couple of conclusions and assumption that were not covered by the results and which were not related to previous literature.
  5. a) Authors wrote “The molecular weight of OVA is 45 kDa, BSA is 68 kDa, Mb is 17 kDa. This proves that the loading effect is also closely related to the structure of the protein.”(line 195-196) The problem with this statement is, that molecular weight and protein structure are two independent things. Moreover authors provide no reference.
  6. b) Authors wrote: “This is consistent with many previous studies” (line 209). Why did the authors not refer to the previous studies?

The same for following statement: “Previous research documents have also shown that different substitution sites and different degrees of substitution also have a greater impact on the activity of chitosan derivatives.” (line 214-215) Please add relevant citations.
